# Prescription Opioid Distribution after the Legalization of Recreational Marijuana in Colorado

**DOI:** 10.3390/ijerph17093251

**Published:** 2020-05-07

**Authors:** Amalie K. Kropp Lopez, Stephanie D. Nichols, Daniel Y. Chung, Daniel E. Kaufman, Kenneth L. McCall, Brian J. Piper

**Affiliations:** 1Department of Medical Education, Geisinger Commonwealth School of Medicine, Scranton, PA 18509, USA; dyc4321@gmail.com (D.Y.C.); dkaufman01@som.geisinger.edu (D.E.K.); bpiper@som.geisinger.edu (B.J.P.); 2Department of Pharmacy Practice, University of New England, Portland, ME 04103, USA; snichols6@une.edu (S.D.N.); kmccall@une.edu (K.L.M.)

**Keywords:** cannabis, fentanyl, Maryland, morphine, oxycodone, opiate, public policy, Utah

## Abstract

There have been dynamic changes in prescription opioid use in the US but the state level policy factors contributing to these are incompletely understood. We examined the association between the legalization of recreational marijuana and prescription opioid distribution in Colorado. Utah and Maryland, two states that had not legalized recreational marijuana, were selected for comparison. Prescription data reported to the Drug Enforcement Administration for nine opioids used for pain (e.g., fentanyl, morphine, hydrocodone, hydromorphone, oxycodone, oxymorphone) and two primarily for opioid use disorder (OUD, methadone and buprenorphine) from 2007 to 2017 were evaluated. Analysis of the interval pre (2007–2012) versus post (2013–2017) marijuana legalization revealed statistically significant decreases for Colorado (*P* < 0.05) and Maryland (*P* < 0.01), but not Utah, for pain medications. There was a larger reduction from 2012 to 2017 in Colorado (–31.5%) than the other states (–14.2% to –23.5%). Colorado had a significantly greater decrease in codeine and oxymorphone than the comparison states. The most prevalent opioids by morphine equivalents were oxycodone and methadone. Due to rapid and pronounced changes in prescription opioid distribution over the past decade, additional study with more states is needed to determine whether cannabis policy was associated with reductions in opioids used for chronic pain.

## 1. Introduction

The US opioid epidemic stems from the early 1990s when the medical community recognized pain as a fifth vital sign [1]. Opioids increased from 148 million prescriptions in 2005 to over 206 million by 2011 [2]. Analysis of the Drug Enforcement Administration’s (DEA) comprehensive dataset identified a marked increase in the total volume of opioids prescribed each year, with a national peak in 2011 [3] followed by declines in most agents [4] with the exception of buprenorphine [5]. Use, and misuse, of opioids has not homogeneously impacted the US. There was a moderate (r = 0.49) correlation between a state’s median age and per capita prescription opioid use [3]. Analysis of the National Health and Nutrition Examination Survey estimated that one-seventh of prescription opioid use was attributable to obesity and associated conditions [6]. States with fewer uninsured people as a result of Medicaid expansion had greater prescription opioid use [7,8]. Use of the OUD pharmacotherapy buprenorphine differed over twenty-fold between the highest (Rhode Island = 2158 morphine mg equivalents (MME)/person) and lowest (North Dakota = 99 MME/person) states [3]. In contrast, use of the high potency prescription opioid fentanyl only showed three-fold state differences [4]. The rate of opioid overdoses among Hispanic people was half that of white people [9]. 

Healthcare providers have the responsibility to treat their patient’s non-cancer pain while also considering nonopioid alternatives. Since California first legalized medical marijuana in 1996, 33 states and the District of Columbia have passed laws broadly legalizing marijuana, either medically or recreationally. As of June 2019, Washington D.C. and ten other states have expanded to condoning recreational marijuana use [10]. With the endorsement of the states, more objective evidence is beginning to emerge that marijuana may be of value to manage chronic pain [11], reduce overdose mortality [12,13], treat opioid withdrawal [14], or decrease opioid prescribing [14,15,16,17,18]. 

Marijuana has a much lower risk of addiction and virtually no overdose danger relative to opioids [19,20,21,22,23]. In January 2017, the National Academies of Sciences, Engineering and Medicine released a peer-reviewed, comprehensive review showing “conclusive evidence” that cannabinoids can be used safely and effectively to treat chronic pain [24]. However, the evidence base was too limited to make a determination regarding whether marijuana could be used to treat addiction to other drugs [24]. Over-three fifths (62%) of Americans are supportive of legalized marijuana for medical purposes, which has doubled from 31% in 2000 [25]. The popularity of marijuana is quickly rising in the fifty years and older population [26]. This demographic may be most likely to experience chronic pain-related conditions and are receptive to the analgesic properties of marijuana [11]. 

To date, there has been little research [15] conducted on the effects of adult-use marijuana laws on opioid distribution. In November 2000, Colorado voters approved Amendment 20, implementing the legalization of medical marijuana. Twelve years later, Colorado approved Amendment 64, legalizing adult-use or recreational marijuana [27]. By January 2014, dispensaries were opened to the public [28]. This report compares medical opioid distribution in Colorado with two states, Utah and Maryland, which had not legalized recreational marijuana. Although the primary emphasis was on opioids used for pain, a secondary objective was to describe prescription opioid use more broadly in these states including those used for OUD treatment.

## 2. Materials and Methods

### 2.1. Procedures

The US Drug Enforcement Administration’s Automation of Report and Consolidated Orders System (ARCOS) was created due to the 1970 Controlled Substances Act to track and publicly report data on controlled substances in Schedules II to III distributed to pharmacies, hospitals, narcotic treatment programs (NTPs, also known as methadone programs), providers, and teaching institutions [29]. ARCOS also reports on opioid distribution by the first-three digits of the zip-code (Appendix A). ARCOS has been used in prior pharmacoepidemiological reports [3,4,5,8,23,30]. Nine opioid medications primarily used for pain: codeine, fentanyl, morphine, hydrocodone, hydromorphone, meperidine, oxycodone, oxymorphone, and tapentadol, and two primarily for opioid use disorder (OUD): methadone and buprenorphine, were obtained by quarter from 2007 to 2017 in Colorado, Utah, and Maryland. Colorado and Maryland had similar demographics in terms of population, home ownership, education, and uninsured rates. Utah was selected as a geographically similar state with some comparable characteristics (Table 1) [31,32,33,34]. Institutional Review Board approval was provided by the University of New England.

### 2.2. Statistical Analysis

All eleven opioids were converted to their oral morphine milligram equivalents (MME). This enabled the agents to be compared despite their differences in relative potency. Oral MME conversions were completed using the following multipliers: oxycodone 1.5, fentanyl 75, morphine 1, hydrocodone 1, hydromorphone 4, oxymorphone 3, tapentadol 0.4, codeine 0.15, meperidine 0.1, methadone 8, and buprenorphine 10 [4]. Heat maps of three-digit zip codes were prepared using QGIS (QGIS.org, Gossau, Switzerland) (Appendix A). Other figures were created with GraphPad Prism, version 8.1. Population data were taken from Statista. T-tests compared pre-marijuana legalization (2007–2012) and post-marijuana legalization (2013–2017) for the nine pain medications and two OUD medications. A *P* < 0.05 was considered statistically significant.

## 3. Results

The main objective of this report was to determine whether recreational marijuana legislation was associated with changes in prescription opioid distribution, particularly for agents primarily used for pain including codeine, fentanyl, hydrocodone, hydromorphone, meperidine, morphine, oxycodone, oxymorphone, and tapentadol. T-tests compared the quarterly distribution pre (2007–2012) and post-marijuana legalization (2013–2017) periods for nine pain medications, expressed as MME and corrected for population. There was a statistically significant reduction for Colorado (*P* = 0.033) but not in Utah (*P* = 0.659). However, Maryland (*P* = 0.007) also showed a decrease during the same interval (Figure 1A). Analgesic opioids peaked in 2011 or 2012 in all three states (Figure 1C). Relative to 2012, opioids for pain decreased by 31.5% in Colorado but only 14.3% in Utah and 23.5% in Maryland. The decline in prescription opioids for pain appeared similar (i.e., there were parallel lines) from 2012 to 2017 for Colorado and Maryland.

Table 2 shows the percent change for 3-digit zip codes from 2012 to 2017. Colorado had a significantly greater decline than Utah for eight of the nine pain opioids: codeine, fentanyl, hydrocodone, hydromorphone, meperidine, morphine, oxycodone, and oxymorphone. However, Maryland had a significantly greater reduction than Colorado for hydrocodone. There were no differences in OUD medications among the states. 

A secondary goal was to describe opioid distribution in these three states. Oxycodone and methadone were the top two opioids by MME in each state (Figure 2A–C). The percent of opioids, by MME, used for pain from 2007 to 2017 showed pronounced differences. Less than two-fifths of opioids were for pain in Maryland (37.89%), versus over three-fifths in Utah (61.00%) and four-fifths in Colorado (79.49%). Analysis of the two OUD agents methadone and buprenorphine, expressed as quarterly MME and corrected for population, identified significant increases for Colorado (*P* = 0.0003) and Maryland (*P* = 0.0001). The decrease in Utah was not significant (*P* = 0.0935, Figure 1B). Further examination of the temporal pattern revealed that Colorado increased OUD treatments from 2007 to 2017 while Utah showed the reverse pattern (Figure 1C) which was driven by declines in methadone (UT: –20.0% versus CO: +21.2%, MD: +7.2%, Figure 2B). 

Figure 3 depicts the percent change from 2012 until 2017 per three-digit zip code for codeine, hydromorphone, and oxymorphone.

## 4. Discussion

The first objective of this report was to determine whether recreational cannabis passage in November 2012 [28] and implementation in 2014 in Colorado, relative to two comparison states that had not yet endorsed recreational marijuana, was associated with increased rates of decline in prescription opioid use. The rationale for this hypothesis was a diverse evidence base resulting from basic [35] and epidemiological data [36] including patient reports [18], drug expenditures [16,17], and overdoses [12]. Table 1 provides some support for this hypothesis with greater reductions in six of the nine studied opioids relative to Utah or Maryland. However, the general temporal pattern in Figure 1, including parallel rates of decline in pain medications in Colorado with Maryland, does not support the hypothesis. Overall, this preliminary report based on only two comparison states provides inconclusive evidence. A recent report that examined the entire country with a more granular dataset of outpatient pharmacies discovered that recreational marijuana law implementation was associated with a 12% reduction in the MME and total days’ supply of prescription opioids [37]. This finding is congruent with a prior study limited to Medicaid [15] indicating that recreational and medical cannabis laws may be, at a population level, beneficial for reversing the earlier excesses in prescription opioids used for pain.

Perhaps a greater strength of this dataset was its detailed description of the changes in prescription opioid use in these three states. There were two key findings. Methadone by MME was far and away the predominant prescription opioid in Maryland. In 2012, methadone accounted for over half of the MME attributed to the eleven studied opioids. In contrast, methadone use was declining in Utah. The number of overdoses involving methadone increased fourteen-fold from 1991 to 2003 in Utah [38]. There was one NTP in 2017 in Maryland for every 74K population, compared with one per 239K in Utah and 244K in Colorado, a three-fold difference. These findings are consistent with and extend upon past OUD treatment research [3,5]. Earlier investigations have examined prescription opioid use patterns in Hawaii, Florida, and Texas [8,38]. Over three-quarters of the total MME in Puerto Rico was distributed by NTPs [8]. Unfortunately, NTPs are not included in most pharmacoepidemiological reports [15,37]. Second, there were appreciable variations in the use of hydrocodone. Hydrocodone and fentanyl in morphine equivalents in 2017 were equivalent in Utah. However, hydrocodone use was only one-third that of fentanyl in Maryland. Utah was ranked second in the US for per capita fentanyl use (1,650 ug) which was almost double that of Maryland (845 ug) or Colorado (894 ug) [4]. Unless there are regional differences in chronic or acute pain conditions, these sizable regional disparities may be inconsistent with evidence-based medicine and warrant further study.

There are some important limitations to this report and future directions. The US DEA’s Automation of Report and Consolidated Orders System (ARCOS) provides an inclusive dataset available for studying opioid distribution [29]. However, ARCOS is unable to account for opioids obtained through illegal or undetected means such as the Dark Web or opioids that cross state or national borders. Additionally, the choice of comparative states may limit generalizability. Colorado and Maryland share similar demographics in terms of population size, home ownership, education, and uninsured rates. However, there is an appreciable difference in the percentage of non-white people living in Colorado (12.7%) versus Maryland (41.0%) (Table 2). In the past two decades (2000–2015), non-Hispanic black people and Hispanics have displayed low mortality rates involving opioid overdose [39]. Future studies may need to take these factors into consideration to account for any differences. Comparison of other states that legalized the recreational use of marijuana during the same time period would also be beneficial. Washington (Initiative 502, 2012) and Oregon (Measure 91, 2014) may serve as comparative states with their similar legalization timelines and demographics [40,41]. Finally, there have been several national and state policy changes which has resulted in reductions in prescription opioid use since the national peak in 2011 [2,3]. Naturalistic studies like this are extremely challenging to interpret as there are multiple demographic and socioeconomic factors which contribute to both prescription opioid and marijuana use [6,7,9]. There may be no single state, or states, that perfectly matches the characteristics of Colorado (Table 2). Future research with more states could employ a multivariate approach in an attempt to statistically account for these differences. A detailed characterization of the temporal changes in these three states, which together have a population of 14.9 million, may have more value than any inference, however tentative, about public policy that can be made with cross-state comparisons. It would also be important and influential to consider the overall health status of Colorado and the comparison states. This would allow for improved comparison of the influence of marijuana legalization.

## 5. Conclusions

In this study, we observed dynamic changes in opioid distribution for eleven opioids used for pain and OUD within Colorado, and two carefully selected comparison states, Utah and Maryland, from 2007 to 2017. Colorado, after legalizing recreational marijuana, had a significant decrease in prescription opioids distributed for pain. The findings from this geographically limited study were challenging to interpret because, while analgesic opioid use was unchanged in Utah, Maryland also had a significant decline. Other national research [15,37] more clearly showed that marijuana policies were associated with reductions in analgesic opioid use. This appears to be an empirically informed public policy strategy which may contribute to reversing the US opioid epidemic. 

## Figures and Tables

**Figure 1 ijerph-17-03251-f001:**
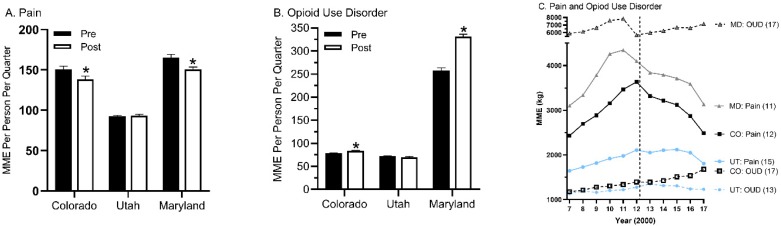
Prescription distribution per quarter (+SEM) as reported to the Drug Enforcement Administration’s Automated Reports and Consolidated Ordering System expressed as morphine mg equivalents (MME) and corrected for population for opioids used for pain (**A**) or Opioid Use Disorder (OUD, (**B**), * *P* < 0.05 versus pre-legalization), or both (**C**) during the period before (2007–2012, designated by a vertical dotted line) and after (2013–2017) recreational marijuana legalization in Colorado (CO) relative to comparison states (MD: Maryland, UT: Utah).

**Figure 2 ijerph-17-03251-f002:**
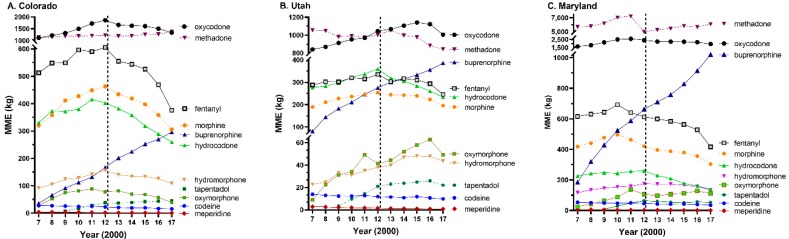
Distribution of each opioid in Colorado (**A**), Utah (**B**), and Maryland (**C**) as reported to the Drug Enforcement Administration’s Automated Reports and Consolidated Ordering System expressed as morphine mg equivalents (MME).

**Figure 3 ijerph-17-03251-f003:**
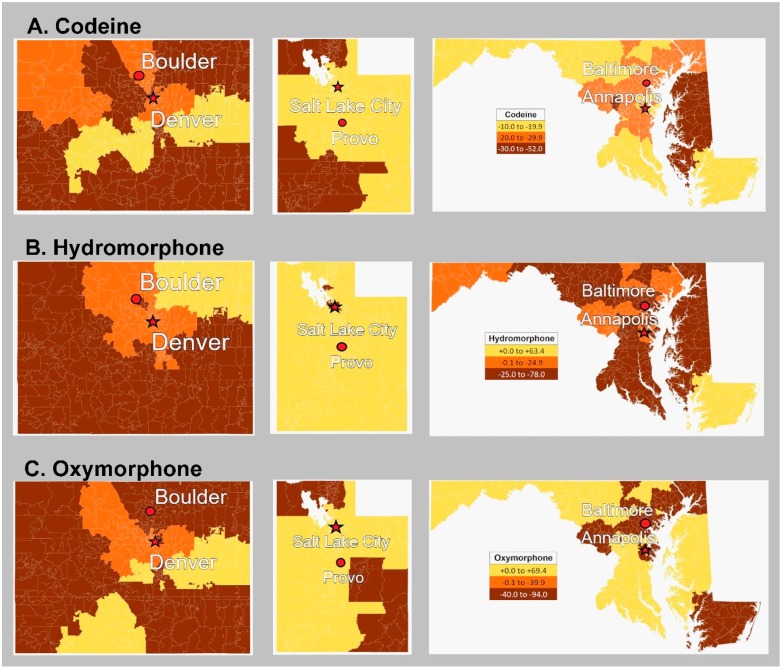
Percent change by three-digit zip code from 2012 to 2017 in codeine (**A**), hydromorphone (**B**), and oxymorphone (**C**) as reported to the Drug Enforcement Administration’s Automated Reports and Consolidated Ordering System in Colorado, Utah, and Maryland, respectively.

**Table 1 ijerph-17-03251-t001:** Demographic comparison of the three states including population, percent uninsured, Medicaid expansion, median household income, percent of population with a bachelor’s degree (BS), and percent of adults with a body mass index ≥ 30 [31,32,33].

Characteristic	Colorado	Utah	Maryland
Population	5,695,564	3,161,105	6,042,718
% uninsured	10.0%	12.0%	8.0%
Medicaid expansion	Yes	No	Yes
Median income	$65,458	$65,325	$78,916
Home ownership	64.7%	69.6%	66.8%
Education (% ≥ BS)	39.4%	32.5%	31.3%
Median age	36.5	30.5	38.5
% Non-white	12.7%	9.1%	41.0%
% Obese	22.6%	25.3%	31.3%

**Table 2 ijerph-17-03251-t002:** Percent change (+SEM) in opioid distribution by three-digit zip code from 2012 to 2017 in Colorado (800–816) relative to two states (840–847, 206–219) without a recreational cannabis policy. Superscript by each opioid designates whether it is primarily used for pain or an opioid use disorder (OUD). *P* < 0.05 versus ^U^ Utah or ^M^ Maryland.

Opioid	Colorado (*N* = 17)	Utah (*N* = 7)	Maryland (*N* = 13)
codeine^Pain^	−30.6 (2.6) ^U,M^	−19.3 (2.1)	−22.6 (1.5)
fentanyl^Pain^	−38.6 (2.2) ^U^	−25.4 (4.0)	−33.6 (2.0)
hydrocodone^Pain^	−35.0 (1.2) ^U,M^	−27.4 (2.2) ^M^	−41.7 (1.8)
hydromorphone^Pain^	−29.3 (6.6) ^U^	+28.2 (9.8) ^M^	−22.2 (3.6)
meperidine^Pain^	−63.4 (2.8) ^U^	−53.4 (2.4) ^M^	−66.9 (3.3)
morphine^Pain^	−35.7 (2.4) ^U^	−22.6 (4.7)	−29.5 (2.8)
oxycodone^Pain^	−27.6 (3.2) ^U^	−6.4 (4.9) ^M^	−21.3 (4.3)
oxymorphone^Pain^	−46.0 (9.6) ^U,M^	+12.4 (9.7)	+31.5 (27.9)
tapentadol^Pain^	+27.9 (18.4) ^M^	+33.4 (23.9)	−19.8 (4.2)
buprenorphine^OUD^	+84.1 (15.0)	+52.0 (15.0)	+102.4 (23.0)
methadone^OUD^	+1.6 (16.5)	−19.1 (8.2)	+85.2 (55.2)

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
