# Peer review of "Prescription Opioid Distribution after the Legalization of Recreational Marijuana in Colorado"

_ijerph, 2020, doi:10.3390/ijerph17093251_

Round 1
Reviewer 1 Report
This is an interesting article that studies the prescription pattern of opioid distribution after legalizing marijuana in Colorado by comparison with two control states, Utah and Maryland.Manuscript is well written and clear.I have one concern over the data collection.Is it possible to collect subject/patient health data? This will help in understanding the change in health pattern in these states. May be overall health status of Colorado state was improved, over the years, which might be independent of marijuana usage.
Author Response
This is an interesting article that studies the prescription pattern of opioid distribution after legalizing marijuana in Colorado by comparison with two control states, Utah and Maryland. Manuscript is well written and clear. I have one concern over the data collection. Is it possible to collect subject/patient health data? This will help in understanding the change in health pattern in these states. May be overall health status of Colorado state was improved, over the years, which might be independent of marijuana usage.
- Added in the limitations section:
- It would also be important and influential to consider the overall health status of Colorado and the comparison states. This would allow for improved comparison of the influence of marijuana legalization.
Reviewer 2 Report
Thank you for requesting my opinion for the review of the manuscript "Prescription Opioid Distribution After the Legalization of Recreational Marijuana in Colorado".
The manuscript is adequately written, and causes little confusion.
There are no major changes to introduce.
Note that the results are presented before the methodology and statistical analysis in the manuscript.
The curves can be simplified to convey a message that is easier to puzzle out.
The limitations should be further substantiated. The choice of comparative states has important limitations to detail too.
I hope this will help.
Issam
Author Response
- Note that the results are presented before the methodology and statistical analysis in the manuscript.
- Noted and fixed from the Molecules to the IJERPH format.
- The curves can be simplified to convey a message that is easier to puzzle out.
- Completed –Figure 1 (A-C), Figure 2 (A-C) were separated.
- The limitations should be further substantiated. The choice of comparative states has important limitations to detail too.
There are some important limitations to this report and future directions. The US DEA’s Automation of Report and Consolidated Orders System (ARCOS) provides an inclusive dataset available for studying opioid distribution [29]. However, ARCOS is unable to account for opioids obtained through illegal or undetected means such as the Dark Web or opioids that cross state or national borders. Additionally, the choice of comparative states may limit generalizability. Colorado and Maryland share similar demographics in terms of population size, home ownership, education, and uninsured rates. However, there is an appreciable difference in the percentage of non-whites living in Colorado (12.7%) versus Maryland (41.0%) (Table 2). In the past two decades (2000-2015), non-Hispanic blacks and Hispanics have displayed low mortality rates involving opioid overdose [40]. Future studies may need to take these factors into consideration to account for any differences. Comparison of other states that legalized the recreational use of marijuana during the same time period would also be beneficial. Washington (Initiative 502, 2012) and Oregon (Measure 91, 2014) may serve as comparative states with their similar legalization timelines and demographics [41, 42].

Reviewer 3 Report
It is a job well done. Some suggestions:
1 What is the contribution of the study?
2. Was the study authorized by a research or ethics committee?
3 could expand the discussion
4. Figure 1, can you divide it into two figures?
Author Response
- What is the contribution of the study?
Requested information added.
- To date, there has been little research [15] conducted on the effects of adult use marijuana laws on opioid distribution. [Lines 72-73]
- The main objective of this report was to determine whether recreational marijuana legislation was associated with changes in prescription opioid distribution, particularly for agents primarily used for pain including codeine, fentanyl, hydrocodone, hydromorphone, meperidine, morphine, oxycodone, oxymorphone, and tapentadol. [Lines 109-112]
- Was the study authorized by a research or ethics committee?
- Institutional Review Board approval was provided by the University of New England. [Lines 93-94]
- could expand the discussion:
- There are some important limitations to this report and future directions. The US DEA’s Automation of Report and Consolidated Orders System (ARCOS) provides an inclusive dataset available for studying opioid distribution [29]. However, ARCOS is unable to account for opioids obtained through illegal or undetected means such as the Dark Web or opioids that cross state or national borders. Additionally, the choice of comparative states may limit generalizability. Colorado and Maryland share similar demographics in terms of population size, home ownership, education, and uninsured rates. However, there is an appreciable difference in the percentage of non-whites living in Colorado (12.7%) versus Maryland (41.0%) (Table 2). In the past two decades (2000-2015), non-Hispanic blacks and Hispanics have displayed low mortality rates involving opioid overdose [40]. Future studies may need to take these factors into consideration to account for any differences. Comparison of other states that legalized the recreational use of marijuana during the same time period would also be beneficial. Washington (Initiative 502, 2012) and Oregon (Measure 91, 2014) may serve as comparative states with their similar legalization timelines and demographics [41, 42]. Finally, there have been several national and state policy changes which has resulted in reductions in prescription opioid use since the national peak in 2011 [2,3]. Naturalistic studies like this are extremely challenging to interpret as there are multiple demographic and socioeconomic factors which contribute to both prescription opioid and marijuana use [6,7,9]. There may be no single state, or states, that perfectly matches the characteristics of Colorado (Table 2). Future research with more states could employ a multivariate approach in an attempt to statistically account for these differences. A detailed characterization of the temporal changes in these three states, which together have a population of 14.9 million, may have more value than any inference, however tentative, about public policy that can be made with cross-state comparisons. It would also be important and influential to consider the overall health status of Colorado and the comparison states. This would allow for improved comparison of the influence of marijuana legalization.
- Figure 1, can you divide it into two figures?
- Completed --This was also noted by reviewer 2 point 2.
